# Task-Centric Application Switching: How and Why Knowledge Workers Switch Software Applications for a Single Task

Amir Jahanlou,* Jo Vermeulen,† Tovi Grossman,‡ Parmit K. Chilana,§ George Fitzmaurice,¶ and Justin Matejka‖

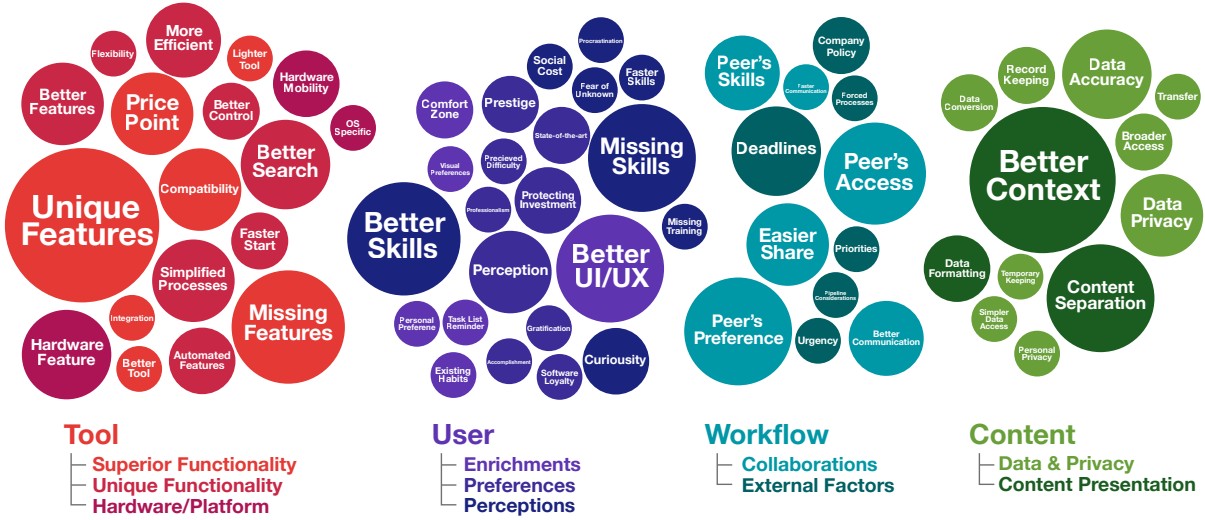

Figure 1: The low-level reasons for application switching captured during this research (colored bubbles) divided into four primary categories (Tool, User, Workflow, and Content), each with sub-categories (listed below the bubbles). The size of a bubble represents the number of responses for that low-level reason on a logarithmic scale. The largest bubble size represents 16 responses, and the smallest size represents a single response.

## ABSTRACT

Knowledge workers often have to switch between multiple software tools to complete *a single task*, which can deter productivity. Previous literature has established the high incidence of application switching that comes with the cost of converting and transferring data or getting distracted and wasting time to re-focus. This research explores *why* knowledge workers deliberately switch between many applications despite potential drawbacks. We interviewed 15 knowledge workers and five product teams to understand why users switch between separate tools to complete tasks. From our results, we synthesize an initial taxonomy of reasons for application switching, illustrate the role of collaboration and external forces, and detail the challenges caused by application switching. We offer design implications for how task-centric application switching can be better supported by promoting multi-tool learning, designing interfaces that enable users to reflect on their application-switching behavior, and application switching analytics.

**Index Terms:** Human-centered computing—Human computer interaction (HCI)—Empirical studies in HCI

---

*Autodesk Research, Simon Fraser University; e-mail: ajahanlou@sfu.ca
†Autodesk Research; e-mail: jo.vermeulen@autodesk.com
‡University of Toronto; e-mail: tovi@dgp.toronto.edu
§Simon Fraser University; e-mail: pchilana@cs.sfu.ca
¶Autodesk Research; e-mail: george.fitzmaurice@autodesk.com
‖Autodesk Research; e-mail: justin.matejka@autodesk.com

## 1 INTRODUCTION

Application switching occurs when a user navigates from one application to another [59] and is common in everyday software use [63]. Research has shown that knowledge workers have eight or more windows open [36] in most situations (78.1%) and make hundreds of switches within a single hour [46, 54]. Such switches come at the cost of launching a new tool, waiting for the tool to load, and in some cases converting and transferring data to another application.

Prior work has explored application switching from multiple causes, such as interruptions [24, 29], multitasking [2], and window switching [63] (more details in sections 2.2 and 2.3). While the literature has documented a high incidence of application switching when interruptions occur [1], this type of switching is often not a conscious decision by the user. A phenomenon that is not as well understood is why users *deliberately* switch applications while completing *a single task* – i.e., when the user switches between applications with the goal of finishing the same task. We define *task* as a specific software-based activity that a user performs to achieve a particular goal, such as editing a video. In this case, a video author might make a conscious decision to switch among different applications for image manipulation (Adobe Photoshop [6]), title creation (Adobe Premiere [7]), sequence editing (Final Cut Pro [11]), or color correction (DaVinci Resolve [19]) to complete the same *task* of creating a video. We characterize the act of switching between multiple applications with the goal of finishing the same task as *task-centric application switching*. While previous works have explored the mechanics of switching behaviors (e.g., the use of shortcuts [61]), the reasons behind these switches and resulting difficulties are open questions, particularly from the perspective of knowledge workers.

Knowledge workers are individuals whose primary function is to create, share, and analyze information [25, 56]. Previous studies [24, 29] have shown that knowledge work is characterized by

multiple ongoing and often disjoint tasks [13]. Although switching between applications provides knowledge workers the opportunity to learn and transfer their knowledge [1], it also increases the cognitive load [2, 54] that could impact overall productivity. Moreover, frequent switching between tools may also prevent users from developing fluency or expertise in any of the individual tools [34]. This paper establishes an initial understanding of the phenomenon of self-initiated [24] task-centric application switching within sample domains of knowledge work. We focus on understanding the reasons for *why* knowledge workers switch between tools and the potential challenges that result from these switches.

We conducted semi-structured interviews with 15 knowledge workers to understand the practices of switching between applications. We also had the unique opportunity to explore this question from the perspective of product teams that design software tools. We conducted group interviews with five product development teams to understand their decisions to embed a new feature, integrate tools within larger applications, and opinions towards learning how to use their software along with other applications. Our findings suggest that users deliberately switch applications due to tool-specific functionality, the need to collaborate with others, company policies, users' attitudes towards using (and learning) feature-rich tools, individual needs for data conversions, or privacy concerns. We synthesized our observations into an initial taxonomy of why knowledge workers switch between different applications for the same task.

Although application switching was perceived to be helpful for the task at hand in some cases, it also introduced challenges that impacted users' productivity. The cost of switching between applications due to data transfer, the time required to re-focus on the task, and the extra cognitive load the users must endure are some of the difficulties that application switching bears. We discuss the implications of how software tools can better support task-centric application switching and help knowledge workers be more productive and efficient in their tasks. In summary, our paper makes the following contributions:

- Initial insights into task-centric application switching, including causes, benefits, and challenges for knowledge workers.

- An initial taxonomy of reasons for application switching.

- Design implications for developers and learning content creators to better support task-centric application switching.

## 2 RELATED WORK

To situate our findings, we draw upon research on software learnability, multitasking, and productivity support tools.

### 2.1 General Software Learnability

Software application users are challenged by featurism, the growing list of application features with each release, and the training that is often focused on a single software tool. Previous HCI research [31, 43, 48] has explored this from the perspective of software learnability. These works have made recommendations for improving the task flow [31], user awareness of the UI [48], improving feature findability [26], and understanding and reducing functionality [20]. Personalizing user interfaces [25] and adding customizability have also been explored. Moreover, contemporary knowledge work is rarely an individual activity [13, 35]. These collaborative environments are typically composed of several arrangements among multiple users and software tools, and collaboration itself causes difficulties with learnability [52]. Many challenges stem from the fact that each software company designs the tool and the training of their respective software, and the practices of knowledge workers—who often utilize several tools together with one another [28, 56]—are not fully considered.

Previous work on supporting individuals with small units of tasks (e.g., document production, email, and communication [28]) has

largely focused on individual software tools and rarely considers the difficulties in utilizing multiple applications to complete a task. To address this need, initiatives in industry and research have resulted in larger software applications instead of multiple small applications that work in concert with each other [27, 44, 45]. While limited functionality tools are only useful for particular conditions, the difficulty of learning complex software [22] and the challenges of working with feature-rich tools [42] make users hesitant to use them.

In this paper, we highlight application switches resulting from challenges of feature findability, collaboration, and the need for appropriate tools that reflect and inform users of their practices. More broadly, our work contributes to a trajectory of research in the HCI community [3, 17, 37] aimed at studying the needs of knowledge workers to better support their practices, procedures, and experiences.

### 2.2 Supporting Users in Multitasking

Previous studies in HCI have explored approaches to support users in multitasking, such as software work interruptions [62] or situations when users lose their flow of work [49]. Much of the multitasking efforts look at "work fragmentation" [60]. In doing so, scholars have explored the role of task management and window switching [34, 63], understanding values that influence the adoption of creativity support tools [55] as well as the inclusion of devices such as mobile phones [58] for work purposes. A notable focus has been on improving application management [41, 53], and in understanding and providing a definition of a unit of task [25, 38, 51]. Furthermore, the utility of larger screens [34], and the benefits of utilizing multiple displays for work [36] as well as some of their difficulties [34] have been explored. Another area of study can be seen in activity-centric computing systems [18], and the analyses of the different window overlapping techniques [40] while simultaneously having a high-level understanding of the process [33].

While prior work demonstrates a few instances in which multitasking has been positive (such as enabling better creativity [47]), most literature on multitasking attempts to address its challenges. Among other difficulties is that for seamless multitasking [54], application switching must happen in near-simultaneous execution [59]. However, switching between applications results in interruption that often taxes the process [28]. It is worth noting that there is a difference between task switching and an interruption, as the former is generally a conscious choice for a more extended period while the latter is usually a temporary shift in attention caused by external factors [1]. Our work complements these previous studies by investigating knowledge workers' specific reasons for task-centric application switching and the challenges they face as a result.

### 2.3 Task Management and Productivity Support Tools

HCI has a long history of exploring productivity measurement and support for software users [3, 13, 17, 37, 48]. Switching between application contexts provides knowledge workers the opportunity to learn and transfer their knowledge [2]. Yet, changing contexts for completing the task might come at a cost [2, 54]. It produces an increased cognitive cost from fragmenting the work and a resumption lag that reduces performance [60]. Previous studies have illustrated how users may opt for sub-optimal techniques because they preferred working with tools they were familiar with rather than exploring (potentially) better alternatives [52]. Such constraints posed by knowledge deficits are a deterrent to long-term productivity. While there have been some efforts in measuring software productivity [49], concrete methods are needed to understand the processes and support users. A suggestion in that direction has been to provide users with project-specific task reminders [24].

Our study reveals insights into the difficulties that occur due to the tools or the users' workflow. We further explore the challenges shaped by individual traits of knowledge workers (such as willing-

Table 1: Overview of the participants in this study representing their domains, age, and years of experience they had in their work.

| P#/PT# | Gender | Position | Age | Exp |
|---|---|---|---|---|
| P1 | Female | Office Assistant | 25-34 | 1-9 |
| P2 | Female | Software Project Manager | 35-44 | 10-19 |
| P3 | Male | 3D Content Creator | 45-54 | 20-29 |
| P4 | Male | Software Developer | 35-44 | 10-19 |
| P5 | Female | Executive Assistant | 35-44 | 10-19 |
| P6 | Female | Community Manager | 25-34 | 1-9 |
| P7 | Female | Architecture Manager | 45-54 | 10-19 |
| P8 | Male | College Manager | 45-54 | 20-29 |
| P9 | Female | Visualization Researcher | 25-34 | 1-9 |
| P10 | Male | Business Manager | 35-44 | 10-19 |
| P11 | Female | Book Author | 45-54 | 10-19 |
| P12 | Female | Research Scientist | 25-34 | 1-9 |
| P13 | Male | Personal Trainer | 35-44 | 10-19 |
| P14 | Female | User Experience Designer | 25-34 | 1-9 |
| P15 | Female | Life Coach | 55-64 | 20-29 |
| PT1-1 | Male | Program Manager | 55-64 | 20-29 |
| PT1-2 | Male | UX Designer | 25-34 | 10-19 |
| PT1-3 | Male | Product Manager | 25-34 | 1-9 |
| PT2 | Male | Product Manager | 45-54 | 10-19 |
| PT3-1 | Male | Product Manager | 35-44 | 10-19 |
| PT3-2 | Male | Product Designer | 45-54 | 10-19 |
| PT4-1 | Male | Product Manager | 35-44 | 10-19 |
| PT4-2 | Female | Design Manager | 25-34 | 10-19 |
| PT4-3 | Male | Product Manager | 35-44 | 10-19 |
| PT4-4 | Male | Product Manager | 35-44 | 10-19 |
| PT5 | Male | UX Manager | 55-64 | 20-29 |

ness to learn new tools or the presence of transferable skills) [50] or their settings (such as working against deadlines and in individual or collaborative environments.) Furthermore, our work complements prior studies by highlighting the software and non-software challenges that result from application switching.

## 3 METHOD

To establish an initial understanding of knowledge workers' processes, successes, challenges, and workarounds in application switching, we conducted semi-structured interviews with knowledge workers from several different domains. Moreover, to understand the perspectives of product managers and interaction designers, we conducted group interviews with members of five product teams (a total of 11 individuals) from a multinational software company. These individuals were actively working on designing new features or integrating previously standalone software tools into a larger application and could share observations of their users' feedback and potential struggles. This paper refers to these teams as PT1–PT5 (product team). All sessions were conducted remotely (via Zoom [64]), recorded, and later transcribed. Sessions lasted 45–60 minutes, and all participants were awarded a $50 gift card.

### 3.1 Recruitment and Participants

**Knowledge Workers:** We interviewed 15 participants (10 female, 5 male), spanning different age groups (P1–P15 in Table 1). We sought to interview individuals who used computer software tools for day-to-day office operations, architectural or UX design, software programming, or scientific research. We tried to obtain a reasonable representation from a variety of domains of knowledge work. Participants were recruited using personal contacts, email advertisements, and snowball sampling over two months in 2022.

**Product Teams:** Gathering perspectives from commercial product teams about software design decisions can be challenging due to privacy and intellectual property concerns. We had the unique opportunity of being the research division of a large software company (over 10,000 employees) with over 100 software products in its portfolio. We contacted several groups working on different products and conducted group interviews to allow people in various roles to add to each other's answers and arrive at a shared understanding. In five sessions, we interviewed 11 individuals (see participants with prefix PT1–PT5 in Table 1 for details on their experience and positions). For 2/5 product teams, we only had access to one team member.

### 3.2 Data Collection and Analysis

**Knowledge Workers:** During the interviews, we asked questions about application switching and their opinion on the benefits or challenges of switching among different tools. We also asked whether they could recall a situation in which switching has been easy or difficult and whether they switched between platforms. Once we established their application switching behavior, we explored the reasons behind their switching and how they felt about these switches. We encouraged our participants to take notes of the switches between applications in the days leading up to the interview sessions. Several people came prepared with notes, and two interviewees also produced diagrams of their software tool usage.

**Product Teams:** We sought to understand the teams' decisions of embedding a feature into their applications, the learnability of the tools, considerations for integrating tools within larger applications, the processes of switching between their applications and other tools, as well as their opinion towards learning how to use their software along with other applications. We further inquired about the role of branding and how users perceive a tool.

All interview transcriptions were coded using the *Atlas.TI* [14] data analysis software. We explored the data from the two groups of participants separately, and the coded data were analyzed to illustrate the different processes, challenges, and workarounds that participants had expressed. We used an inductive analysis, and axial coding [23] approach to explore the themes around our main research question. To ensure the validity of the coded data, the primary author performed the first open coding pass and consulted with other researchers to discuss and develop an initial list of codes. Upon completing the first phase, researchers collectively examined the emerging themes and finalized the coding scheme.

## 4 REASONS FOR APPLICATION SWITCHING

Based on our analysis, we were able to classify the reasons for task-centric application switching into four broad categories: *tool*, *content*, *workflow*, and *user*. We synthesized these reasons into an initial taxonomy and explored each category in detail based on the perceived impact of application switching on participants. This taxonomy of reasons for application switching can help future scholars and software developers provide systems by which users' performances can be measured and improved in specific categories.

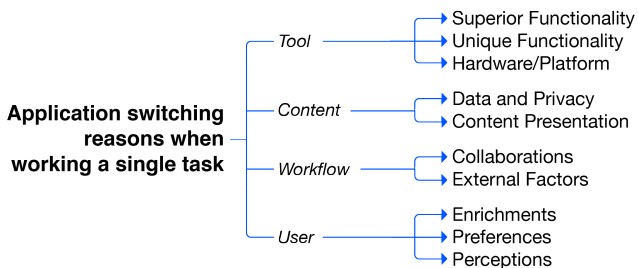

Figure 2: The initial taxonomy of task-centric application switching with categories and subcategories.

### 4.1 Taxonomy of Reasons for Task-Centric Application Switching

Our data analysis initially revealed 67 reasons for task-centric application switching. We did another coding pass to work towards clustering these reasons into categories and subcategories (Fig. 2). These categories include tool, content, workflow, and user. We synthesized an initial taxonomy of reasons for task-centric application switching based on these categories. Fig. 2 represents this breakdown and the various categories and subcategories of reasons.

### 4.2 Tool-Specific Reasons

In this section, we explore the reasons for application switching that pertain to the nature of software tools, such as the superiority of one tool's features, the uniqueness of its features, or the mere availability of an application on a specific platform.

**Superior Functionality:** Our participants worked in different domains with varying levels of software tool training. Some (such as P3 or P8) had spent years earning mastery of their applications. Others had to learn many aspects of the software work on the job. Regardless of the skill level, all participants had to continually work against deadlines, and for that, finding superior functionality was a concern. For example, P2 working as a project manager, described how they preferred switching to a tool over starting a new document in the current one simply because the other tool would fire up faster. This view was echoed by PT4: *"When we integrate a tool, we add new capabilities. That's great, but we also should work on making the interaction seamless and keep the tool fast enough. If the software becomes too heavy, we lose users."* Another participant—who worked on 3D content creation—explained how they switch between different tools to maintain multiple records for easier sharing:

*"A lot of inter-company discussions are happening on Slack. But I like sending emails. Because it is easier to find data. So, if I say something on Slack, I back it up in an email as well. I guess Outlook and email are easier to find information. They do a better job of information retrieval." (P3)*

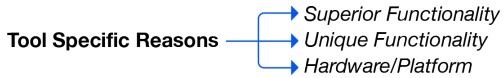

❝ *We use Slack for communicating in our office, but it doesn't allow us to send large files. So, in the middle of a conversation, I have to go to email or the [Google] Drive. (P2)*

Figure 3: The high-level reasons for application switching that relate to the nature of the tools.

**Unique Functionality:** Beyond superior functionality, participants also spoke of many situations where they had to switch to a different application because of its unique functionality. An example was a user wishing to edit an SVG figure before using it in a presentation in Google Slides [30], which they could only do with a particular software tool (Adobe Illustrator [5]). This was the only software that offered this functionality (to the participant's knowledge.) While this approach results in further application switching, PT2 viewed it as something positive: *The constant demand by users for adding new features is only making the tool complicated. Our tool is very simple, it's on the web, doesn't need installation or maintenance. It's supposed to help users move just one step forward. They can do the rest in a different tool." (PT2)*

**Hardware/Platform:** While participants spoke of various superior or unique functionality in their tools, there were also situations when they had to switch to different hardware or operating system. We learned that some of these switches had developed organically, and the users had no recollection of how they had adopted such practices. Similar to a previous study on practitioner values for the

adoption of creativity support tools [55], some participants deliberately chose to switch to a different hardware platform (e.g., P13 for the more accessible keyboard on their phone, P6 using their phone for taking a quick photo, or P15 to get up from their desk.) Some of these switches were also related to the specifics of hardware:

*"When I want to do [a] freehand drawing, I use my tablet for quick ideation. Now, my laptop has a touch screen that I can draw. But it's not very convenient. So I go back to my tablet instead." (P4)*

While some of these were a conscious choice, some had other reasons. P8, for instance, expressed how they occasionally do sound editing and have to switch to a different operating system (Mac OS) because their sound editing tool is only available for Mac. Others constantly navigated back and forth between hardware and preferred to switch when absolutely necessary:

*"I usually get the notifications on my phone, but then I go to the desktop application. I use the phone just to get notified that something requires my attention. Beyond that, it rarely happens that I use my phone for actual work." (P9)*

A summary of the tool-related high-level reasons for application switching can be found in Fig. 3.

### 4.3 User-Specific Reasons

In this section, we explore application switching reasons that stem from the individual differences between the users. Examples of such differences can be found in personal preferences, the training they had received, and the perceptions of the difficulty (or the ease) of using a tool. The user-specific reasons can be summarized in Fig. 4).

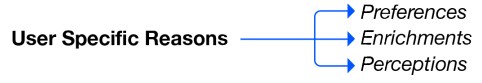

❝ *It would be easier if we could do everything right within Unity. But, none of us knows how to do it, so we end up creating the UI mockups once in Figma and then in Unity. (P14)*

Figure 4: The high-level reasons for application switching that relate to the characteristics of the users.

#### 4.3.1 Subcategories of User-Specific Reasons

**Individual Preferences:** Individual preferences factored in users' application switching practices. Participants such as P15 simply did not want to step outside their *comfort zone*. They would switch to a different, more familiar application, despite knowing there is a way to do that task in their current tool. As found in previous literature [21], they could not be "bothered" with learning the feature. P7, for instance, suggested that they prioritize productivity: *"I take a longer route just because I know how to do that. It's the frustration of having to search for the answer, as opposed to knowing you gonna get there."* Our product teams spoke about how users' habits impact their application use which was not always a positive outcome:

*"We have users who use our tool for years, now, there's also a new generation of customers. They use iPads and touch phones. For them, they expect the UI to work similarly. Now, we have to align our UX with their preferences as well." (PT5)*

Others, such as P8, had reservations regarding application switching, particularly because of the user experience. This was an area brought up by product teams as well:

*"If we integrate a tool to our system, it should follow what users expect. So, that is an area that we spend a lot of time on aligning hotkeys and mouse-keyboard interactions." (PT5)*

**User Enrichment:** Beyond the users' preferences, another major reason was learnability and user enrichment. As seen in previous work [21, 39, 48], for most participants, such as P12, the tool was viewed as mere means of tackling a task, and the mastery of the

software was not a high priority: *"I want to know [how] to accomplish what I am trying to accomplish. I am not interested in learning many new tools. I want to be able to Google something and just find the answer."* Our product teams somewhat reluctantly agreed with such assessments of feature convolution:

*"We invest countless hours integrating various features. Most users don't use these and would prefer to switch to an application with fewer [features] where they feel more comfortable. This way, they finish their work, instead of learning better features." (PT4)*

The disinterest in learning new tools was also impacted by the number of applications. P1, for instance, said: *"...in our onboarding training, they were showing the new hires the hundreds of applications that we have in the company, like for HR or for expenses or for other things. It's mind-blowing, can you imagine how a new hire deals with that? Wow!"* Multiple other participants (P4, P5, P9, or P11) spoke of their lack of knowledge in using a software tool. P1, for instance, mentioned not receiving the right training: *"So, right now, I can't send a sheet automatically to any of three services we communicate with. The guy who knows how to do it doesn't have the time to teach me. So, I end up going to three tools one by one"* Our product teams had mixed beliefs about this.

On the other hand, we also spoke to participants who welcomed personal enrichment through learning new things. For instance, we spoke to a participant who had taken the radical route of developing a tool to reduce the amount of application switching by combining multiple functions of different tools:

*"..., you have a conversation, then you realize it needs to be recorded. Sometimes it becomes the project requirement. I needed a tool to transfer casual chats into actual project details." (P4)*

**Users' Perceptions:** We should make a further distinction of user preferences based on what they perceive. An unexpected finding was when we spoke to P1, who suggested that they switched between two applications merely to have the second software used *somewhere in the pipeline*. Their rationale was that the other software is viewed as a more prestigious application and helps them look more professional. Another example is that we would generally assume that users might wish for better software applications. However, participants such as P9 challenged this notion by wishing that they could protect their time investment: *"If I put the time to learn something, then I'd like to be able to use it again. I feel the energy that went into [learning] it, shouldn't go to waste."* We found an interesting confirmation of this view from product teams:

*"..., our users have invested in learning the highly specialized software. The integration is somewhat 'cheapening' their efforts. I've heard users say, 'now everyone can use this'." (PT1)*

While more experienced participants view this as a way to capitalize on what they have already learned and focus on the task at hand, others, such as P14, viewed it as a bottleneck for new and better ways: *"I don't know the reason. But I think it comes from their background or places in careers. New people are willing to make changes. The more experienced wish to stay within their comfort zone. They force old ways and old tools." (P14)*

The challenges of learning new tools were a more prominent situation for experienced users. PT4, for instance, spoke of an interesting aspect: *"Users now [after integration] have to learn the UI of a new software only to be able to do what they already have been using. For new users, this is okay. For existing ones, they have to first un-learn and then re-learn the new interface." (PT4)*

The initial encounter with the software tools shaped some of the users' perceptions. P11 suggested that: *"If my first interaction has been rewarding, I prefer to go back to that tool because it gives me the feeling of success."* Product teams also spoke of such behavior by their users. PT4, for instance, reiterated *"Some tools are useful for some edge cases. Yet, the user goes there once, gets used to it, if they like [it], they use it for everything."*

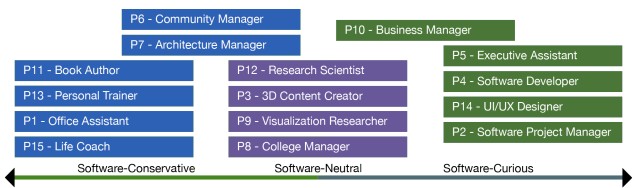

Figure 5: Three groups of users emerged in this research.

### 4.3.2 Groups of Users and Application Switching Behaviors

As we conducted interviews and later in our analysis, we saw general "groups" of users appear. We organized these as broad groups of software-conservative, software-neutral, and software-curious users. This categorization was influenced by various parameters such as their willingness to explore new software applications, ability to balance deadlines with onboarding new tools, or desire to support and train other individuals.

Fig. 5 illustrates the summary of these groups across our participants. Our *software-curious* participants usually had a genuine interest in learning new applications. They often welcomed the challenges of learning or onboarding new software and were seen as technical people in their workplaces. They would express their feeling towards applications or switching using phrases such as *"I'll try it on my own first," "In my free time, I'll play around with it,"* or *"I look for a better solution."* From this category of users, we met P5, who expressed: *"If I have to learn a new tool and switch to it, I don't mind it. I am very tech-savvy, so I don't mind the trouble. I see the issues as a new opportunity for learning and improving myself."*

At the other end, we saw the group of *software-conservative* users. This group behaves similarly to what is described in the *paradox of the active user* [21]. These users are mainly motivated by productivity and have little interest in learning software or exploring new features or software tools. Participants in this group often expressed their feeling towards applications or switching using phrases such as *"I don't know how to do that," "It seems very difficult,"* or *"I can't be bothered."* Unlike the curious group that saw the benefits of mastering software applications, this group viewed the software as an extra task and was concerned about how they had to overcome the challenges of the software on top of their daily work. P15, for instance, expressed: *"I can't try new tools. I'm happy to do things the way I do. I love learning new stuff, but tech isn't one of them."*

As we went through our analysis, we noticed a few participants that could not be categorized as either curious or conservative. This group was neither intimidated by the software switching nor enthusiastic about the tools. They would express their feeling towards applications or switches using phrases such as *"If I have to," "I wouldn't try on my own"* or *"If someone shows me how to do it."* They would simply use the software applications to move forward to the next level. If they had to, they could easily get started with new tools and try to learn their intricacies, but that would be when they clearly felt a necessity. We categorized these individuals as *software-neutral* users that can be represented by P9: *"I told myself you have to relax about technology. It's a [matter of] necessity. If it is not absolutely necessary, I will not learn it."*

### 4.4 Workflow Specific Reasons

This section explores the reasons for application switching driven by users' workflows. While the workflows of individuals are vastly different, we observed distinct trends within the contexts of collaboration and external factors (Fig. 6).

**Collaborations:** Collaboration was perhaps the most influential aspect. In line with prior findings [55], we observed multiple instances in which users would radically change their processes

**Workflow Specific Reasons** → *Collaborations* / *External Factors*

❝ *We use Discord for all of our communications. Then there are clients we work with who only have Slack, so I have to move between these two to maintain conversations. (P10)*

Figure 6: The high-level reasons for application switching that are based on the various workflows.

**Content Specific Reasons** → *Data and Privacy* / *Data Presentation*

❝ *I type up a contract in Word, but when I want to share it with our contractors, I want to make sure they can't change it. So, I convert it to .pdf and then add a password in Adobe [Reader]. (P2)*

Figure 7: The high-level reasons for application switching that are based on the content.

depending on whether they worked with others or on their own. A participant, for example, expressed:

*"I have to make distinct decisions about my own use versus my students. Whenever we have to add a new tool for our students, I'm rather hesitant. I want them to focus on what they have to learn as opposed to focus on their tool instead." (P8)*

PT1, also encouraged software tool integration for a similar reason: *"The larger application means everyone in the pipeline, designer, engineer, managers, or anyone else shares the project by opening the same project within the same software tool. Then, everyone is talking about the same thing."*

Beyond the necessities of switching (such as using a different tool in the latter stage of a pipeline), we also observed several examples of switching directly related to stakeholders' use of different tools. P9, for instance, suggested: *"I am for adding a new tool. These are little tools, so their learning curve is pretty straightforward. My colleagues don't want to make such changes. We end up staying with the challenging workflow."* While some product teams believed in bringing everything into one platform to simplify the viewing context, this was not necessarily true from users' perspective:

*"In our office, the biggest challenge is the perception of integration, not the actual technology. We have tools that provide all [the] different functionalities required, but our colleagues don't like them or don't believe that they can do everything. So, they end up [switching to] alternative tools." (P5)*

Similar resistance to change, particularly when it required training, was observed by other participants as well. P14, for instance, noted: *"Sometimes, if you invest two hours this week, you'll save 20 hours by the end of the month. But most of my colleagues, almost none of them, will ever do that."* This view was also echoed by PT3: *"With each release, we ship an entirely new set of documentation and training material. Most users, unfortunately, skip those."*

Our product teams spoke of how they rely on the communities as an *extension* to the software. Our participants, such as P10, had a similar view: *"I make quick decisions about whether I like something or not. It's based on usability and the community. Can I jump into Google and quickly get answers without getting frustrated?" (P10)*

**External Factors:** Beyond colleagues' and stakeholders' preferences, some factors were built into the processes. Among others, regarding dealing with deadlines, many participants spoke of having to switch between applications simply because of time. P7, for instance, suggested that they had faced a situation where they switched (from one CAD software to another) to add some details. Still, they were unsure if the switch was necessary because it was only doable in one software or if they lacked the knowledge to use the first software. Working against a clock also meant that participants ended up on certain paths. P1, for instance, suggested that: *"Most often, a lot of things are happening fast, and we are busy. So, I end up just doing things manually that I know will finish the task."*

### 4.5 Content-Specific Reasons

Finally, we observed patterns in how participants dealt with their data and content (summarized in Fig. 7). A lot of these were affected by the need for privacy. P6, for instance, suggested: *"When I'm done in Word, I save the file as .pdf. I can then open it in Adobe [Reader]. It has really great features for setting various levels of access for*

*contractors."* Other participants, such as P4, also echoed the need to switch between applications for various privacy reasons. In their view, the lack of privacy settings might result in them eventually migrating entirely from one tool to another. Beyond privacy, participants also spoke of the ways that they had to separate their content. Some participants spoke of how they use different software solutions for different aspects of their operations that enabled a better degree of content separations:

*"For the goto market planning and ideation and the documentation, we use Miro. For me, everything is there. Everything is altogether over there. If I need something in UX, I go to Figma. This separation is good. This allows me to give access to the right people in sales, marketing, etc." (P14)*

Particularly in collaborative settings, we encountered examples when participants would switch from their current tool simply to view things the same way their stakeholders would. P3, who works on 3D content, suggested that depending on the next person in the pipeline, they use different applications to ensure that the data can be opened and manipulated correctly. Product teams also mentioned the viewing context as an important driver for merging applications. PT1, for instance, was pleasantly surprised with a similar situation: *"The visual context is something we didn't expect. Previously, users had to separate part of the data and execute a series of tasks with different tools. After the integration, they get a visual context of all the content at once that makes it more intuitive."*

## 5 CHALLENGES CAUSED BY SWITCHING APPLICATIONS

Although our participants had many reasons for switching between tools, and some willingly switched for more successful workflows, most faced substantive challenges.

### 5.1 Challenges of Learning the Many Tools of a Pipeline

Most of our participants had to employ multiple unrelated software tools. P5, for instance, brought a diagram (Fig. 8) that represented nearly 15 steps and multiple tools required to complete a contract. Others, such as P12—who worked as a research scientist—sometimes did not even know if they were using the right tool: *"I don't like having this burden of search to see if I can do something in that software, as opposed to first identifying the software that I need. I prefer knowing that I am in the right tool."*

Another challenge was the increasing feature-richness of tools, which created skill gaps for users. Participants spoke of how they had received initial training on using a software tool, yet they were unaware of the features added in each new release. PT1 also spoke of similar challenges: *"With each release, we have to add many new features. If we only fix the bugs and make the current tool better, we risk users dismissing it because we haven't made much innovation."*

The lack of awareness of features makes users' skills obsolete. The increasing richness of software applications also meant that the tools had become so complex that participants wished to access lighter, limited versions. P8, for instance, suggested that they would welcome an alternative to their 3D tool. Similarly, P7 expressed: *"You know, the software grew because we needed functionality. The problem is it's harder to learn. Needs much more investment in terms of time... there are functions that I'd never use."* As observed in prior studies [42], for many participants, trying to employ new

tools or features required looking to others for help. This approach comes with its own challenges. P4, for instance, articulated: *"Right now we have a challenge that many things are just the knowledge of an individual. The process shouldn't be based on the knowledge of one person, rather the software that is helping the user."*

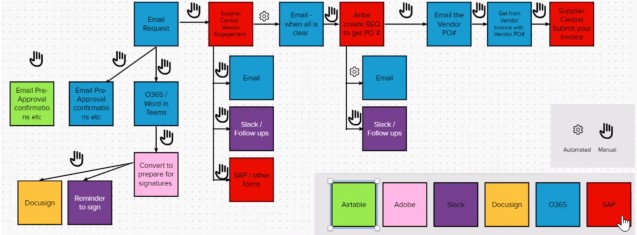

Figure 8: An example of switching among multiple applications illustrated by P5 to complete the task of getting a contract signed. The user has to navigate back and forth among numerous application.

## 5.2 Missing Interoperability Between File Formats

Although the technical capabilities of tools were important drivers of application switching, we observed that application switching also happened because users needed more customizability, better viewing of the content (e.g., in a simpler interface), or the ability to isolate specific parts of the content. P9, working on scientific data visualization, for instance, spoke of how their tools did not communicate with one another using standard file formats. Another participant, who worked on 3D content authoring, expressed:

*"I work with people who use MotionBuilder [16]. So, I have to open files in Motion Builder because I need to make sure I see what they see. This way, I can be sure that we both are talking about the same thing. We have that common frame of reference. But if there is another team that uses Maya, then I test the file on Maya." (P3)*

The challenges of viewing data in different representation modes were not unique to 3D authoring. In fact, our participants recounted many situations where they needed to isolate their content to view them within a different interface (e.g., pieces of a text being edited in a separate view, isolating parts of graphics, reviewing sections of numeric data). These content transfers often resulted in multiple copies of the files, the need for saving and re-opening the data, and occasionally feeling lost between multiple copies:

*"There's a copy of the files on the server, but not everyone keeps that up to date. So, I maintain a spreadsheet that is manually updated, and I compare it to the one on the server. So, I have two files that have to be manually compared" (P7)*

Participants complained that their software tools did not allow such content separations or that the user interfaces were rather crowded. In essence, these are capabilities currently built into the tool that don't benefit users due to sub-optimal interaction design. Some of these could be addressed by better discoverability of the features, others by designing better user interfaces.

## 5.3 Understanding and Measuring Productivity

Our participants also spoke of the difficulties of keeping track of many tools and the numerous times they switched between them throughout the day. P1, for instance, suggested: *"Because there are many tools, I can't remember which one of use for what. So, I have written a list for myself. I switch there to find out what tool is used for what. It's like a list of tricks."* In our participants' view, the sheer number of tools and the ongoing introduction of new ones were deterrents to productivity. P7, for instance, advocated for reducing the number of communication tools, while P3 suggested: *"Sometimes it [switching] is a pain because I have to go back to another software application. It'd be nice to have a merged application because*

*you could do everything in one package. It breaks up your rhythm of working."* Despite this interest, participants were wary of the customizability trade-offs that might come with larger applications:

*"[Microsoft] Teams is a good example. Even though it's available to us, we don't use it because it doesn't allow customization. It's everything for everybody, so it's not all that customizable. Then some of the things that we really need, we need IT intervention, which doesn't make sense. The workflow shouldn't involve IT." (P8)*

P6 viewed customizability from a different perspective: *"I think using many small applications gives me more flexibility. They are separate, but [I] can switch between them more flexibly. Many small apps give me a bird's eye view."* In summary, our participants, spoke of challenges such as:

- Software learnability is often concerned with specific tools; the entire pipeline of software applications needed to finish a task is not considered. Moreover, new releases of tools require re-training.

- Participants were challenged with the lack of interoperability among applications and the constant need to change file formats and open data files in separate applications only to view their content in a certain way.

- Users complained about having to make many switches without any real way of reflecting or visualizing their processes. They advocated for reducing the number of tools while enabling further customizability.

While, for many, these application switches developed organically, and participants were perhaps impervious to them, some, such as P8, were aware of the costs: *"I have that high context-switching penalty. It would have to be a seamless integration for me to use integrated tools; otherwise, I prefer switching to the standalone one."*

## 6 POSITIVE ASPECTS OF APPLICATION SWITCHING

Although knowledge workers faced several challenges when switching applications for the same task, they also discussed many positive aspects that facilitated switching through which they could extend their current workspace and seamlessly work with the same data on different applications.

One area our participants appreciated about switching applications was data availability and access. P2, for instance, spoke highly of the default applications installed on Apple devices and the availability of the AirDrop [8] feature and cloud storage in iCloud [12] across different devices: *"Notes [9], for instance, when I enter something in it on my phone, it's automatically on my laptop. Also, sometimes, when I copy text, Siri [10] makes suggestions for what I should do with that. Like if it's for a calendar entry, maps, or text messaging someone."* Some participants also spoke of the benefits of using single sign-in applications that host their data on the cloud. For these users, the backend of the cloud provided the necessary functions for accessing their data, and the single sign-in nature meant that they had immediate access:

*". . . , like if I have a presentation, I sometimes copy tables directly from Google Sheets into a presentation. It's just there, and it's very convenient. I click on the button, I'm automatically logged in, and my tables get connected." (P8)*

Enabling such single-login cloud servers was also a major effort for our product teams. PT1, for instance, suggested: *"By integrating our tools, we also moved to the cloud. That means login, storage, and transfer are activities that the user no longer needs. They can now only focus on their actual job."*

Another approach that facilitated switching was live data integration. These could be seen when users chained a series of files from different applications while still keeping them interconnected. P3, for instance, spoke of how they used the file *"reference"* feature in Maya [15] while others were still working on those files. Changes

applied to such files are immediately reflected on their scene. P14 spoke of the benefits of this approach in motion graphics: *"Sometimes, when I make simple animations, I make a file in [Adobe] Photoshop [6] and then open its layers in [Adobe] After Effects [4]. I can then go back to Photoshop and make changes. They show up immediately in After Effects.*

Finally, participants sometimes switched between applications to remind themselves of the tasks ahead. P2, for instance, talked about how launching a tool (that was very resource-intensive) meant they could take a little break. P9 had a similar take over doing repetitious tasks: *"If I want to turn my mind off, I go to a task that is mainly legwork. I take a break like that."* We also spoke to participants who benefited from multiple software tools to divide their work and segregate their content. P14, for instance, divided the creative and marketing work using two different software while P4 suggested: *"I use each software specific to one application. Like one IDE for Java, etc. This helps me have a division between my work and benefit from knowing when one work ends and the next starts."*

## 7 DISCUSSION AND FUTURE WORK

One of our key contributions of this paper is in providing an initial taxonomy of why users switch applications for the same task, highlighting key themes of events such as individual preferences or the requirement of collaborative work. Our participants mentioned many reasons why they switched to another tool while completing the same task (Fig. 1), such as individual skills, preferences, team constraints, or the nature of the work (e.g., having an important deadline). Although many users expressed that application switches were costly and detrimental to productivity, these users acknowledged that using multiple software tools was integral to how knowledge workers performed tasks.

In fact, a key lesson learned from this study is that **task-centric application switching appears to be here to stay.** This has important implications for researchers and software vendors who need to recognize that users *will* indeed switch between software tools, likely to those outside of a single vendor's suite of applications or even tools on different hardware platforms. If the software is designed with this behavior in mind, perhaps some of the challenges our participants experienced can be alleviated. With this understanding, we have three main takeaways for researchers, application designers, and learning content developers:

### 7.1 Development of Multi-Tool Learning Materials

Many application-switching challenges can be traced back to the difficulties in learning and understanding how multiple tools work together. Learning material focused on using a single product will continue to play an important role. However, given the prevalence of using multiple programs (often from different vendors) to complete a task, we encourage developers and learning content creators to further focus on creating learning materials that show how a piece of software can be used as a part of a larger, diverse, pipeline of tools for completing particular tasks. Such efforts can further enable users to focus on their tasks instead of learning the tools and increase knowledge workers' productivity. In doing so, it is also important to ensure these efforts consider the specific user. To provide customized training specific to each user, the system can rely on the user's experience, background, and previous training, rather than using generic practices that may only apply to some users.

### 7.2 Enabling Self-Reflection on Switching Behaviors

During this research, we met participants who faced difficulty understanding (let alone navigating) the many software tools they had to utilize to complete a single task. Several individuals expressed how they had to keep notes of tools and their functionality to remember which software they should use for which part of the task. We also came across a participant who had developed a reminder system to

inform them of the next software they should use in their work. This can be attributed to the greater number of back-and-forth switches between the different tools that make application switching nearly invisible for most users. As we learned from the product managers, new tool additions are happening organically over periods of time, and users need to be made aware of these. A suggestion for future work is to develop tools that can track and personalize individual users' workflows. Currently, a limited number of tools (e.g., [32,57]) are able to track different applications and contextualize them for the user. Developing applications that can document individual workflows can enable a degree of self-reflection. We can draw upon previous self-reflection approaches introduced in HCI and visualization research. For example, future work can explore techniques to support presenting each step of the process within a broader context of the entire pipeline to help users understand how different stages relate to one another. The key goal here would be for knowledge workers to not only better understand their tool use but also actively reflect on each of the steps. An example of such self-reflection could be users viewing their processes in deadline-driven settings and assessing their effectiveness.

### 7.3 Application Switching Behavior Analytics

Many software vendors rely on using analytics to understand feature usage and gauge performance. However, this is usually only focused on a single tool. Our findings suggest the need to expand the scope of analytics to include application switching behaviors to develop a more complete picture of how users accomplish tasks. For example, future research can explore tracking the frequency of different types of task-centric application switching based on our taxonomy. This could provide insights into where feature usage is dropping off and being compensated with another tool. This information could be useful for UX teams to improve the usability of such features. It can also help product teams in setting benchmarks, for example, aiming for less than 10% of switches that occur due to user-specific reasons.

## 8 LIMITATIONS

Our study relied on self-reports of 15 participants and interviews with five product teams to provide a first exploration in this space. While our taxonomy is based on one interpretation of the data, it provides a starting point for isolating and tackling specific challenging areas. Once task-centric application switching can be tracked effectively, future research could build on our results to explore task-centric application switching using a data-driven approach with a larger number of users and over a larger period of time. This could also allow researchers to discover how prevalent certain categories of task-centric application switching in our taxonomy are for different tasks and software applications, and in different contexts. We studied task-centric application switching with a broad range of knowledge workers. Future work could expand on this by investigating application switching of individuals in particular domains and capture domain-specific practices.

## 9 CONCLUSION

Our work explored task-centric application switching, contributing new insights into why knowledge workers deliberately switch applications when completing *a single task*. Our findings demonstrate the processes, benefits, and challenges of application switching devised and exercised across many information domains. As a growing number of knowledge workers use multiple software tools in concert with each other, application switching is here to stay. Partitioning complex software features to support users' learnability is crucial to enhancing overall productivity. Our results also underscore the importance of a more nuanced understanding in HCI of the interoperability among different applications and how knowledge workers leverage both limited-feature standalone tools and larger feature-rich software applications for the same task.

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
