# OpenReview forum: "Task-Centric Application Switching: How and Why Knowledge Workers Switch Software Applications for a Single Task"
_graphicsinterface.org/Graphics_Interface/2023/Conference — GI 2023_

### Official Review · Reviewer_qRve · 2023-01-01
**scope to big and unclear definition of a single task**

**Rating:** 5
**Confidence:** 4

**Review:**

This paper presents an interview study with knowledge workers to investigate why they switch software applications in their daily tasks. It tackles a very important problem and better understanding workers’ behavior would allow us to build better tools. In general, this paper is well motivated and well written.

However, there are a few concerns that prevent me from rating the paper higher. First, the paper seems to aim too high by studying all kinds of knowledge workers. There exists much diversity in different industries/sectors, which stretches the paper too thin. Second, the interviewees are highly prone to the tech/software industry. Further, all the product teams are from the same company. The above limitations will generate bias on the results, which constraints the generalizability. The authors may wish to tamp down their claims or focus on just the software industry, which I think is still a critical problem domain; however, this results in some of the interviewees not qualified. Third, the definition of “a single task” is unclear. What is the scope? For example, adjusting the lighting of an image can be viewed as a task, and crafting a poster can be also viewed as a task which includes the image adjustment task. This ambiguity will produce many compound variables when interviewing with the users, because they may have their own definition of “a single task.” I didn’t see how this definition is clarified with the interviewees. This makes the results in this paper less generalizable.

Minor points: the authors should include a full list of interview questions somewhere on the paper.

Given the above drawbacks, I think this paper needs a round of careful revisions that may be difficult to achieve within the review cycle.

---

### Official Review · Reviewer_HXz8 · 2023-01-12
**An excellent exploration on the specific case of use in modern day productivity process for knowledge workers**

**Rating:** 6
**Confidence:** 4

**Review:**

The paper describes explorative research on why knowledge workers determine which software to use to complete their tasks and how the decision impacts productivity using interviews with knowledge workers. The results revealed primitive reasons for application switching, and offer design implications for task-centric applications.
The introduction was well written and expanded great views and motivations of why multiple applications are being used to achieve the same task by knowledge workers. Overall an interesting topic, and I believe is a good degree of the stage to claim as an exploratory study which can also be useful for commercial products to benchmark and take advantage of to provide better productivity.

Below are my other comments:
- It would be great to have terms defined, for example, task, goal, mission, etc. There are multiple and different scopes of achievement that can be defined. If switching one messenger platform to another to talk with a collaborator, then is sending a single message a task?
- In Table 1. if Age was collected as a range, how was the average calculated? wouldn't the median be more representative? Was this because the distance was equivalent for all options?
- It would be great to have the actual count/frequency of the responses for each reason, in addition to the graphical bubble size

---

### Official Review · Reviewer_NMYz · 2023-01-17
**Interesting and timely topic. Thoughtful coding and analyses from the interviews. Results are rich and useful too.**

**Rating:** 8
**Confidence:** 5

**Review:**

The paper presents an interview study that involved 15 knowledge workers and 5 teams to understand the reasons and mechanisms why workers switch between software applications in work. Results show that multiple aspects of considerations are pertinent, which can be tool-, user-, workflow, and/or content-specific. Overall, I found the paper well-motivated and well-written. It's easy to follow and read. The method is well-planned and executed, and the results are meaningful. It could be a bit counter-intuitive to see the results that people switched applications for reasons beyond just features or functionality at a first glance. The authors did a good job in organizing and presenting the results with appropriate levels of abstraction and context, and the findings just make a good sense.

The paper has a few minor issues in my read. One is that the term "task" was not defined or used in a very consistent way. There could be a difference between switching between applications to achieve a specific goal, versus switching between apps to achieve different goals. The former seems conceptually closer to what people normally refer to as a task. But many of the cases and examples for application switching in the paper seems to be closer to the later. The design implications seemed relatively lightweight with respect to the rich results obtained. I didn't see too many ideas emerging out of the results for design. The implications and contributions to the design space certainly can still be further improved.

---

### Meta-Review · Area_Chair_5FDT · 2023-01-18

**Recommendation:** 7
**Confidence:** 4

**Metareview:**

The paper has received quality reviews from three reviewers. The reviewers shared the sense that the the paper is well-motivated and well-written. The work was also considered to be tackling an important problem domain, and the results were considered to provide better understanding that can be applied toward building better tools.

With that said, the three reviewers have different overall ratings and areas of concern with the paper. Scoping and defining terms like "task" is one common recommendation made across three reviewers that can be improved when preparing the final version if the paper is accepted. There're one to two other issues raised by reviewers, but all appeared to be rather minor and not critical to the level that may affect the contribution of the work.